# Feasibility and First Experiences from an Online Kidney School for Patients with Chronic Kidney Disease

**DOI:** 10.3390/ijerph20010864

**Published:** 2023-01-03

**Authors:** Anders Nikolai Ørsted Schultz, Stefan Rowald Petersen, Tove Fibieger, Jan Dominik Kampmann, Eithne Hayes Bauer

**Affiliations:** 1Department of Internal Medicine, University Hospital of Southern Denmark, 6200 Aabenraa, Denmark; 2Department of Regional Health Research, University of Southern Denmark, 5230 Odense, Denmark; 3Center for Innovative Medical Technology, Odense University Hospital, 5000 Odense, Denmark

**Keywords:** telemedicine, chronic kidney disease, patient education, telehealth

## Abstract

*Objectives:* To examine patients’ and relatives’ experiences with participation in an online kidney school (OKS) and its influence on their choice of treatment modality; furthermore, to report on healthcare professionals’ (HCPs) first experiences with OKS. *Methods*: A mixed-methods design with parallel data collection involving two questionnaires for participants, including patients and relatives and a focus group discussion (FGD) with HCPs. *Results*: The OKS was feasible, and overall, patients and relatives were satisfied. Participation in the OKS increased the percentage of those who felt ready to make a decision regarding treatment. One over-arching theme (*evolvement of the online kidney school over time*) and three sub-themes (*concerns and perceived barriers, facilitators,* and *benefits and future possibilities*) emerged from the FGD. *Conclusions:* The OKS proved feasible, was well-accepted, and increased participants’ abilities to choose a preferred treatment modality. HCPs displayed initial concerns regarding the quality of the OKS and worried about the practicality of conducting the OKS. They experienced a feeling of something being lost. However, over time, HCPs developed strategies to tackle initial concerns and discovered that patients were more capable of participating than they had anticipated.

## 1. Introduction

Chronic kidney disease (CKD) affects more than 10% of the population worldwide [1]. CKD can be divided into five stages according to glomerular filtration rate (GFR). Stage 4 (GFR 15–29 mL/min/1.72 m^2^) refers to “severely decreased” and stage 5 (GFR < 15 mL/min/1.72 m^2^) refers to “kidney failure” [2]. In the final stages of the disease, patients will need either dialysis, kidney transplantation, or maximum conservative management. Dialysis types can be divided into different categories, with home peritoneal dialysis (PD), home haemodialysis (HHD), and conventional in-centre haemodialysis (HD) being the main treatment options [3]. In 2021, approximately 346 per 1,000,000 people received dialysis treatment in Denmark [4]. Patient education can improve understanding of the disease and treatment options [5]. Additionally, CKD patient education programs have been shown to have several beneficial effects, including the delay of dialysis initiation, improved survival on dialysis, and positive effects on psychological well-being [5]. In a survey of patients with end-stage renal disease, the patients’ abilities to identify chronic comorbidities were associated with lower mortality, more so than for patients unable to identify chronic comorbidities [6]. As such, patient education is an important part of the care and treatment of patients with CKD. At the Hospital of Southern Denmark, a kidney school for patients—mainly with CKD stage 4—and their relatives has existed for more than 15 years. The kidney school consists of three classes of three hours over three days teaching patients with CKD and their relatives about end-stage renal disease and treatment options. It includes lectures from a nephrologist, a dialysis nurse, and a dietician, followed by group lunches after each day to facilitate social interaction among the participants and course teachers.

On 11 March 2020, the World Health Organization (WHO) declared the outbreak of the novel coronavirus SARS-CoV-2 (COVID-19) a global pandemic [7]. Subsequently, Denmark closed major parts of the public sector [8]. The kidney school at the Hospital of Southern Denmark was closed for an unknown period of time, resulting in longer waiting lists for patients and their relatives. In response to this unprecedented challenge, an online kidney school (OKS) was established. Online or Mobile health (mHealth), as it is broadly referred to in the literature [9], might well be more relevant in the future of patient education [10]. As the OKS was a novel solution borne from an extraordinary situation, it warranted further investigation to explore the experiences and lessons learned from those involved, in order to provide valuable insights for others interested in mHealth education.

In this paper, we report on the feasibility and first experiences from the OKS. The specific aims of the study were to: (1) establish if an OKS was feasible, (2) examine patients’ and relatives’ experiences of the OKS and how participation in the OKS influenced choice of treatment modality, and (3) investigate healthcare professionals’ (HCPs) experiences and perception of the OKS.

## 2. Materials and Methods

This study was designed over a short period of time in response to the clinical situation during the COVID-19 pandemic in the Nephrology Department of the Hospital of Southern Denmark, which necessitated a pragmatic approach to methods and data collection.

A convergent mixed-methods design was employed, with parallel data collection, followed by analyses of quantitative and qualitative data separately. Data were then compared and integration occurred at the interpretation and reporting level [11]. A mixed-methods design allowed for the examination of different aspects of the OKS, expanding our knowledge of the first experiences of patients, relatives, and HCPs with OKS. Guided by Malterud, we conducted a focus group discussion (FGD) exploring the different experiences of the HCPs involved in the OKS [12]. Qualitative data were analysed using a thematic analysis [13]. We assessed patients’ and relatives’ experiences and knowledge acquisition using questionnaires and reported data using percentages, medians, and interquartile ranges (IQR).

### 2.1. Setting and Participants

The OKS was conducted at the Hospital of Southern Denmark. Patients were eligible if they lived within the catchment area of the hospital, had CKD stage 4, and were on the waiting list for the physical kidney school. The relatives of these patients participated alongside the patients. Patients were excluded if they did not have an internet connection or a device to access the OKS.

The material for the OKS was developed in collaboration with a nephrologist, dialysis nurses, a dietician, a graphic designer, and a research nurse with webinar experience. The original material from the physical kidney school was converted to an online format based loosely on the experiential health information-processing model for web-based patient education [14]. By using this model, we endeavoured to create material that could be taught in a collaborative manner where learning styles and interaction are weighted highly, rather than merely providing knowledge transformation. Furthermore, we incorporated Salmon et al.’s., five-stage model for learning online, where the teacher moderates, facilitates, and teaches participants in the virtual environment [15]. This was to ensure that the teachers of the OKS were aware of their multifaceted role in facilitating learning in a virtual environment in the OKS [15]. The level of interaction between the participants and teachers was, thus, dependent on at what stage of the model they found themselves. In the early stages of the model, technology takes the helm, but as one progresses from one stage to another, more interaction can take place, technology dominates less, and more learning can occur [15]. The OKS consists of a three-hour synchronous class using the Cisco Webex platform, which is taught by a nephrologist and a dialysis nurse who have received instructions in hosting webinars. The nephrologist and dialysis nurse taught the online class from a conference room at the hospital equipped with video conferencing equipment. Interaction between the course teachers and participants during the OKS was achieved through individual questions and pauses for questions, both spoken and through the chat function. The webinar was combined with 1–1 telephonic advice from a dietitian and four asynchronous animated short films on treatment options produced by the dialysis nurses for the OKS.

The participants (patients and relatives) could join the OKS from their homes together or separately, thus, facilitating the participation of relatives living in the home, further away, or indeed, on another continent. Patients received a link via secure mail with instructions for use, which, if so desired, they could share with a relative. Telephonic support was provided, if necessary.

Patients who declined participation in the OKS were placed on a waiting list for individual counselling with a dialysis nurse and a dietitian, when possible.

### 2.2. Data Collection

#### 2.2.1. Assessment of Feasibility

Feasibility was assessed by: (1) the number of patients who were willing to participate in the OKS and (2) the number of patients who were able to connect to the system and participate in the OKS.

#### 2.2.2. Qualitative Data Collection

All five HCPs who were involved in the OKS were purposively selected to participate in the FGD, all of whom agreed to participate. The FGD was held in the hospital, at a time convenient to the HCPs. The first author moderated the FGD, and the fourth author was present as an observer taking notes. Both the first author and fourth authors were medical doctors and PhD students. The interview guide was developed by the first, fourth, and last author, but not piloted (see Appendix A). The fourth author had a professional relationship with the participating HCPs and, therefore, did not moderate the FGD. The fourth author was not involved in the OKS at any stage. The FGD was audio-recorded and the data were transcribed verbatim. Transcripts were not provided to participants for verification. The first, second, and last authors analysed the data using an inductive thematic analysis following the steps outlined by Braun and Clark [13]. The transcripts were read and re-read so as to become familiar with the data. Initial coding was performed independently by all three authors involved in the qualitative data analysis, followed by a meeting to align and categorise the codes and initial themes. The themes were reviewed and the original data revisited to validate the themes [13]. The themes are presented with supporting citations. The consolidated criteria for reporting qualitative research (COREQ) was employed for reporting the qualitative results of this study [16].

#### 2.2.3. Quantitative Data Collection

Two questionnaires—a locally adapted Kidney Knowledge Questionnaire (KKQ) adapted from Devins et al.’s Kidney Disease Questionnaire [17] and the Telehealth Usability Questionnaire (TUQ) [18]—were employed in this study. The participants in the quantitative part of this study comprised patients and relatives, and henceforth, are referred to as participants/responders. The patients and relatives taking part in the OKS received links to an online survey, which they could complete anonymously. KKQs were distributed prior to and post-participation in the OKS to assess the participants’ knowledge of CKD and treatment preferences. The TUQ was distributed post-participation to evaluate the participants’ experiences of the quality of the user interface and the quality of the telehealth interaction and service during the OKS [18].

#### 2.2.4. Kidney Knowledge Questionnaire

Adapted from Devins et al.’s Kidney Disease Questionnaire, the questionnaire was translated to Danish and underwent substantial revision [17]. The fourth author translated the questionnaire and then presented it to the kidney school providers, who recommended amendments to align the questions to the curriculum. The fourth author amended the questionnaire accordingly and presented it to the kidney school providers, who ensured that the amendments were aligned with the curriculum. An English version of the adapted KKQ is attached as Appendix B. The KKQ consists of nine knowledge questions plus one question regarding the preferred treatment option in case of kidney failure. All knowledge questions were multiple choice with four possible answers, of which only one was correct.

#### 2.2.5. Telehealth Usability Questionnaire

The TUQ is a validated questionnaire consisting of 21 questions, on a 7-point Likert scale (1 = strongly disagree and 7 = strongly agree), covering five domains; Usefulness, Ease of Use, Effectiveness, Reliability, and Satisfaction [13].

The Danish translation of the TUQ was carried out according to the International Society of Pharmacoeconomics and Outcomes Research (ISPOR) guidelines for Good Practice for the Translational and Cultural Adaptation Process for Patient-Reported Outcomes Measures, following the first seven steps [19]. The ISPOR guidelines include: (1) preparation, (2) forward translation, (3) reconciliation, (4) back translation, (5) back translation review, (6) harmonization, (7) cognitive debriefing, (8) review of the cognitive debriefing results and finalization, (9) proofreading, and (10) final report [14]. The translation has not yet been validated for internal consistency.

### 2.3. Statistics

The demographics of the questionnaire responders are described by sex, presented in numbers and percentages, and by age, presented as medians and IQRs. The results of the KKQ are presented as a percentage of total responders divided into the following three categories: correct answers, incorrect answers, and don’t know.

The results of the TUQ are presented with medians and IQRs, as the responses did not follow normal distribution—see Appendix C. All analyses were performed using R version 4.0.2, R Foundation for Statistical Computing, Vienna, Austria.

## 3. Results

### 3.1. Feasibility

From 21 January 2021 to 6 April 2022, the OKS was held eight times. Forty-four patients were invited, and a total of 30 completed the OKS. The median number of patients participating was 3.5, ranging from two to six.

Of the 14 patients who did not participate, the reasons for not doing so were as follows: six were unable due to technical issues, four refused any form of kidney school, three preferred to wait for the physical kidney school, and one for an unknown reason. Relatives also participated, although no data were collected on how many.

### 3.2. Qualitative Results

Only one FGD could be performed, as the number of HCPs involved in the set-up of the OKS was limited to five people from the Nephrology Department. However, the FGD included all five HCPs, all of whom contributed to the discussion with varying experiences and perceptions. The FGD lasted for 45 min, and the HCPs demonstrated ease in conversing with one another, which led to a high level of interaction and discussion. Some of the HCPs were directly involved in teaching the OKS, and others were not, but all were involved in developing the concept of the OKS. This ensured variation within the group, along with differing age and occupation, which provided the basis for differing opinions in the FGD. One main overarching theme emerged from the data *(evolvement of the online kidney school over time)*, along with three sub-themes (*concerns and perceived barriers, facilitators, benefits and future possibilities*). The over-arching theme and sub-themes are presented below. As this was a single-site study, the demographics of the participating HCPs are not included in order to ensure participant anonymity.

#### 3.2.1. Over-Arching Theme: Evolution of the Online Kidney School over Time

The HCPs discussed how the concept of the OKS developed and evolved over time, which led them to question the goals of both the physical kidney school and the OKS. Despite initial differing opinions, they all concluded that knowledge acquisition in itself was not the main goal, but that patients developing an understanding of their kidney disease and treatment options was.

*“It’s not that they have to be able to answer specific questions. They have to have an understanding of what kidney disease means and what treatment options there are … generally speaking”*.(Participant no. 3)

#### 3.2.2. Sub-Themes

##### Concerns and Perceived Barriers

Some HCPs described a fear of delivering inferior quality when teaching the OKS. Their initial concerns rested on whether or not the OKS would be inferior to the physical school and whether or not the two forms of teaching were comparable.

*“…I can see it as a disadvantage, if we find out that they (ed. Participants) don’t get just as much out of being online, as they do when they are physically present”*.(Participant no. 1)

They pointed out that the OKS has no live lecture from a dietician and would entail less interaction with participants. However, some also described their experiences of the OKS, in which the participants were satisfied.

*“And the oldest we’ve had online, he was eighty-nine and he was just so; ”well, I’ll manage this!” and then he managed to get online, though we had neither camera nor sound from him and he wasn’t able to ask any questions, but he could take part … like, he could take part …He could hear XX’s lesson and he told (ed. us) that he got a lot out of it. It wasn’t a problem for him”*.(Participant no. 1)

Some of the HCPs expressed concerns that were of a technical nature and were related to their ability, and that of participants, to use the equipment, especially when HCPS reported previous experiences with online meetings that did not go well. They also shared concerns over what type of technology to use and the availability of the same for patients. Some HCPs feared that the OKS would be time-consuming, even though the online course was shorter than the physical course, as they pointed out.


*“That was a bigger worry (..), in relation to whether they (ed. Participants) were able to connect. On the other hand, there were just many things that happened at once because so much was converted to online and there were many ways of doing things, so the possibility was there. But of course, what system should we choose? And was it available to them at home? And how many of them (ed. Participants) even had the opportunity to get onto the system from their home?”*
(Participant no. 4)

A few of the HCPs expressed psychological barriers; a feeling of something being lost with the OKS in comparison to the physical kidney school, which was difficult for them to define. They discussed this in relation to a lack of control of the participants’ understanding of imparted knowledge from the OKS.


*“You don’t have quite the same sense of who you’re speaking to and how it’s being received… Not the same as sitting in a room with them face-to-face...”*
(Participant no. 3)

##### Facilitators

Most of the HCPs discussed how gaining experience with the set-up of the OKS and, for those who taught the OKS, developing strategies with teaching the OKS helped them gain confidence, leading to a positive milieu for participants. They experienced that this led to an increase in participants’ willingness to take part in the OKS, fewer technical problems, and more room for content development.

*“And we’ve had some classes that were really small because people backed out anyway, until I realised that if I call them and say ‘now I have a class, are you on?’ and they said ‘nah, I don’t really know’ and then I could take care of it there and then and say ‘Listen here, I think you should try. We have good experience with it and (..) if you don’t get through, well then, we’ll find another solution for you’ and then it was like they were kind of ’well, OK then, I’ll try”*.(Participant no. 1)

Furthermore, they described how their attitude to the OKS changed as their teaching methods developed. Their doubts about what was possible to teach online diminished. Interaction levels during the online lessons increased and participants asked questions more frequently.

*“And some of what we found out, was that, it was good to hold some short breaks (…) and also ask, (..) ‘What do you think about this?’ To kind of get people included because otherwise you’ll just sort of sit there at home in your living room and hide and maybe not participate in the same way as you would have done. (…) So, we’re getting smarter each time we hold it”*.(Participant no. 1)

All HCPs described how patients were more capable of participating in the OKS than was initially anticipated. Most patients displayed a willingness to try. When technical issues arose, resourceful relatives facilitated participation in the OKS.

*”You got a bit of a surprise when you called them and afterwards (ed., found) that there were actually a lot who were interested in it. (…). But it was an eye opener when you called around and found out that (..) some simply said that ‘I have a relative that can help’”*.(Participant no. 4)

#### 3.2.3. Benefits and Future Possibilities for the Kidney School

All HCPs discussed whether the OKS should continue following the pandemic and how conducting the OKS had been a revelation for them in relation to new possibilities. All agreed that the OKS should continue in some form in the future. Furthermore, dietician advice, which was not offered online, became possible as a telephonic consultation. They discussed how digital inclusion would create a possibility for reaching different patient populations, such as patients in employment and patients who declined the offer to participate in the physical kidney school. Furthermore, the HCPs discussed including relatives who otherwise would not have the opportunity to participate, such as relatives living abroad, which they perceived as an avenue for future development.

*“Yeah, so it’s actually opened up for something new, because most often, it’s a wife or a husband who joins in. Now it was other relatives, you know, like children or the like, that logged on from other places”*.(Participant no. 4)

Other future possibilities that the HCPs discussed involved incorporating new teaching methods, such as asynchronous videos, that could be viewed at patients’ convenience, and hybrid solutions where the physical school and OKS could take place together. All displayed a willingness to develop new ways of teaching the kidney school.


*“We could be more flexible as well, also if there is a patient who doesn’t have the opportunity to come in but would like to have it online, then you can hook them up to the physical, can’t you?”*
(Participant no. 2)

### 3.3. Quantitative Results

In total, forty-eight participants (including both patients and relatives) responded to the KKQ prior to participation in the OKS, while 24 responded afterwards. Twenty-one responded to the TUQ. For demographics, see Table 1.

#### 3.3.1. Kidney Knowledge Questionnaire

Forty-eight participants completed 422 knowledge questions prior to participation, leaving 10 questions unanswered. Of these, 60% responded Don’t know, 21% responded correctly, and 18% responded incorrectly. Post-participation, 24 participants completed 202 questions, leaving 14 questions unanswered. Following participation, only 22% responded Don’t know, while 42% responded correctly, and 36% responded incorrectly.

When questioned regarding treatment preferences, 54.3% responded Not enough information prior to participation, decreasing to 27.3% post-participation. The question regarding peritoneal dialysis displayed the biggest change, from 6.5% prior to OKS participation to 40.9% post-participation, (see Figure 1).

#### 3.3.2. Telehealth Usability Questionnaire

Twenty-one participants initiated, and 17 completed the questionnaire. Six responded Not relevant to question number 17. Ease of Use and Effectiveness scored highest, both with a median of 6.0, and with an IQR of 2.0 and 3.0, respectively. The remaining three domains had a median of 5.0, with Reliability showing the biggest IQR of 4.0, while Satisfaction and Usefulness each had an IQR of 3.0, as displayed in Table 2.

### 3.4. Merging Interpretation of Results

#### 3.4.1. Kidney Knowledge Questionnaire and Qualitative Results

Following participation in the OKS, a large number of participants who had previously responded “don’t know” to the knowledge questions now ventured to respond to them. Double this amount responded more correctly than before and double as many responded incorrectly. This may lead one to question whether the goals of the OKS were achieved. However, from the qualitative data, our understanding of the goals with the OKS was expanded (”It’s not that they have to be able to answer specific questions” FGD (Participant no. 3)); rather, the aim was to gain some understanding of their illness and start the process of reflecting on suitable treatment options. HCPs displayed concerns regarding inferior quality, lack of interaction, and dietician involvement in the OKS. Despite this, there was a positive and large increase in participants who felt confident enough to reply to knowledge questions and choose a treatment modality following participation in the OKS.

#### 3.4.2. Telehealth Usability Questionnaire and Qualitative Results

Initially, HCPs feared that patients might not be able to use the equipment (“Cos that was a bigger worry for you [HCPs], in relation to whether they [the patients] were able to [use the equipment]” FGD (Participant no. 4)). However, the results from the TUQ were mainly positive, with Ease of Use and Effectiveness scoring highest among the participants. These results mirror the results from the FGD, where technical issues were only a problem in the beginning. Moreover, HCPs experienced that, over time, their teaching skills evolved, leading to more interaction: “So, we’re getting smarter each time we hold it.” FGD (Participant no. 1). This mirrors positive TUQ responses regarding satisfaction. However, in the FGD, HCPs still described the feeling of something being lost, which may be reflected in question (15): I think the visits provided over the telehealth system are the same as in-person visits, where two participants responded neutrally and 16 were divided equally between positive and negative responses.

#### 3.4.3. Benefits and Future Possibilities for the Kidney School

In the FGD, it became evident that HCPs believed that the OKS was a success and should continue in some form. This was especially apparent in relation to flexibility and the possibility of reaching different patient populations, including relatives. Although the TUQ does not explicitly examine this, positive responses regarding overall satisfaction and positive responses to question (2) (Telehealth saves me time traveling to a hospital or specialist clinic) can indicate a positive potential for future patient education via an OKS.

## 4. Discussion

Our study shows that conducting an OKS is feasible, and overall, patients and their relatives were satisfied with the quality of the user interface and the quality of the telehealth interaction and service during the OKS. HCPs described an initial concern regarding the inferior quality of online education, which might be well grounded. A review of patient education for CKD patients’ material readability, which included online material, found the language to be too difficult [20]. Few studies have evaluated the quality of online education or mHealth education in general for CKD patients, but a scoping review regarding mHealth education for CKD is currently being conducted [21]. A recent Cochrane review comparing mHealth education for heart failure to usual care showed no differences in knowledge acquisition [22]. Evidence is, however, uncertain regarding self-efficacy, self-care, and health-related quality of life [22]. In terms of knowledge acquisition, the OKS improved participants’ willingness to respond to knowledge questions in the kidney knowledge questionnaire, but still with some uncertainty. Regarding treatment modality, the OKS increased the percentage of participants who felt ready to make a decision regarding treatment and gain sufficient information in order to change their preferred treatment modality to peritoneal dialysis; however, 27.3% responded that they did not have enough information to make a decision yet.

This finding is considerably lower than that of *Easom* et al. in their home-run telemedicine patient education study [23]. However, in their study, participants received three educational classes of eight hours, whereas in our study participants received only one class of three hours [23]. When comparing the results following the first class of three hours to ours, the results are similar; *Easom* et al. displayed an increase in knowledge acquisition and ability to choose treatment modality in 30% of the telemedicine group and 26% in the face-to-face groups, whereas our increase was 27% [23]. This increase in knowledge, ability to choose, and preference for peritoneal dialysis is in line with findings from other studies on mHealth education for patients with CKD [24,25].

In relation to concerns and perceived barriers, HCPS described technical barriers, psychological barriers, and the feeling of something being lost when using video. Interestingly, all of the aforementioned have also been described in a qualitative study investigating CKD patients, their care partners, and clinicians’ perceptions of telehealth visits (video or phone) compared to face-to-face visits [26]. The feeling of something being lost was also reflected on in the results from the TUQ, where many of the participants did not think that the video solution was the same as an in-person visit. Despite the feeling of something being lost, the remaining responses from the TUQ did not resonate in relation to the HCPs’ concerns, most of which changed over time, as the HCPs gained more experience, highlighting the overarching theme of evolvement.

Educating HCPs further to teach online and develop strategies for increasing participant and teacher interaction may help to mitigate perceived barriers and reduce concerns about a gap in the quality between OKS and physical kidney schools. Furthermore, regarding mHealth education, dietary coaching intervention (including phone calls and text messages) has been found to be well accepted among CKD patients [1]. In summary, we found that overall, HCPs experienced that patients and relatives were more satisfied with the OKS than they expected. Likewise, in relation to technical barriers, the HCPs described that patients and relatives were more capable of and willing to participant than they had anticipated. Discrepancies between patients’ and HCPs’ perceptions of mHealth are also displayed in a survey from Israel examining satisfaction with video consultations [27]. While patients were highly satisfied, HCP satisfaction was considerably lower, most probably due to technical and administrative issues [27]. In line with our findings, a review on patient education for patients with CKD suggested that innovative approaches, such as digital media and the inclusion of family and caregivers, might help overcome barriers for patient education [5]. A recent survey of mHealth for patients with CKD concluded that many individuals with CKD use the Internet and smartphones and are interested in using mHealth in the future [28]. This provides a platform for growth, as HCPs have a desire to provide flexible solutions to meet the educational needs of patients with end-stage CKD and their families. Families and caregivers are known to be key factors for support and changing behaviour in patients with CKD [29,30]; however, many caregivers lack knowledge, feel unprepared, and do not receive sufficient support [31]. Therefore, providing technical support to patients and relatives should be considered a factor relative to the success of online patient education.

### 4.1. Limitations

Our paper has several limitations. As this study was conducted in response to a clinical problem during the COVID-19 pandemic, the design and methodology were pragmatically chosen. The choice of method could have benefitted from further rigour. However, we adhered to the recommended reporting guidelines.

Due to the small group of HCPs involved in the OKS, only one FGD was performed, and the number of participants was limited; moreover, the interview guide was not pilot tested. Furthermore, the OKS was held at one hospital, thus, reducing the generalisability and transferability of the results. However, several of our findings are mirrored in the literature.

Due to the anonymization procedure, we were unable to differentiate whether the patients or their relatives answered the questionnaires. Relatives may experience the OKS differently and investigation regarding the different perceptions of relatives and patients may yield important information. In relation to the questionnaires employed in this study, the Danish TUQ translation has not yet been validated. The KKQ, inspired by Devins et al.’s Kidney Disease Questionnaire [17], is likewise yet to be validated. The sample size was limited and only approximately half of the participants who completed the KKQ prior to participating in the OKS responded afterwards. Patient preferences may not necessarily match the treatment they will ultimately receive. Moreover, the overall goal of the kidney school is to help start a process of choosing treatment modality—this process was not measurable but will need future research.

### 4.2. Clinical Implications

Implications for future implementation and assessment of the OKS. Our suggestions for clinical implications are tentative based on the analysed data and available literature. Our findings suggest that resourceful family members were of great help when needed. Furthermore, the OKS presented an opportunity for relatives living further away from patients and abroad to participate in OKS. Hence, an OKS may have the possibility of reaching different groups of patients and/or relatives who can increase their knowledge, receive support and play a more active and supportive role in the informal care of patients, as noted by the HCPs in our FGD. This finding is worthy of further investigation.

There is a need for validation of both questionnaires used in this study. Furthermore, the OKS was not compared to the physical kidney school; however, the survey is ongoing and results on comparability will be reported later.

## 5. Conclusions

The OKS was feasible, well-accepted, and increased patient’s and relatives’ abilities to choose a preferred treatment modality. The HCPs had initial concerns regarding the quality of the OKS, worried about the feasibility of conducting the OKS, and experienced a feeling of something being lost. However, they discovered that the patients and relatives were more capable of participating than they had anticipated, and through experience, they developed new teaching methods. The OKS progressed from a necessity during the COVID-19 pandemic to a new possibility for including different patient groups and relatives. Further research should focus on how to optimise the OKS in the future and investigate the different experiences and learning aspects amongst patients and relatives.

## Figures and Tables

**Figure 1 ijerph-20-00864-f001:**
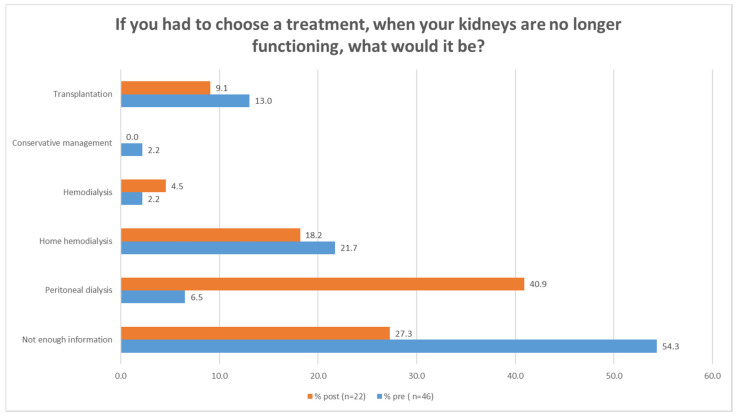
Patients’ preferred treatment option in case of kidney failure, prior to and after participation in the online kidney school.

**Table 1 ijerph-20-00864-t001:** Demographics of patient responders.

Demographics	KKQ		TUQ
	Pre (n = 48)	Post (n = 24)	Post (n = 21)
Male % (n) Female % (n)	69% (33) 31% (15)	79% (19) 21% (5)	76% (16) 24% (5)
Age median [IQR] (n)	69 [10.0] (45)	69 [15.0] (21)	69 [10.75] (18)
Completed questionnaire - Alone % (n) - With relative % (n)	65% (31) 35% (17)	57% (13) 43% (10)	55% (11) 45% (9)

**Table 2 ijerph-20-00864-t002:** Telehealth Usability Questionnaire (TUQ) reponses from participants (both patients and relatives) after participation in the online kidney school.

Statements ^1^	Domain	Strongly Agree	Agree	Somewhat Agree	Neither Agree Nor Disagree	Somewhat Disagree	Disagree	Strongly Disagree	Not Relevant	Number of Responders	Median ^2^ [IQR]	Domain Median ^2^ [IQR]
		Number of Responders										
1	Usefulness	4	5	3	6	1	1	1	0	21	5.0 [2.0]	5.0 [2.0]
2		7	4	2	4	0	3	1	0	21	6.0 [3.0]	
3		4	7	3	2	1	3	1	0	21	6.0 [2.0]	
4	Ease of Use	4	4	5	3	2	1	1	0	20	5.0 [2.0]	6.0 [2.0]
5		3	9	2	1	2	1	1	0	19	6.0 [1.5]	
6		4	8	0	2	1	3	1	0	19	6.0 [2.5]	
7		4	8	2	2	1	1	1	0	19	6.0 [1.5]	
8		4	7	0	2	1	3	1	0	18	6.0 [2.75]	
9		3	5	4	2	2	1	1	0	18	5.0 [2.0]	
10	Effectiveness	2	7	3	1	2	1	2	0	18	5.5 [2.75]	6.0 [3.0]
11		5	4	2	1	2	3	1	0	18	5.5 [3.75]	
12		5	4	5	0	0	3	1	0	18	5.5 [1.75]	
13		4	7	2	0	2	2	1	0	18	6.0 [2.5]	
14		4	4	2	2	1	4	1	0	18	5.0 [3.75]	
15	Reliability	1	3	4	2	1	6	1	0	18	4.0 [3.0]	5.0 [4.0]
16		1	6	6	1	0	3	1	0	18	5.0 [1.75]	
17		0	2	2	3	0	2	2	6	17	4.0 [3.0]	
18	Satisfaction	4	5	1	0	3	3	1	0	17	6.0 [3.0]	5.0 [3.0]
19		2	5	4	2	0	2	2	0	17	5.0 [2.0]	
20		3	5	3	2	0	2	2	0	17	5.0 [2.0]	
21		3	6	3	1	0	3	1	0	17	6.0 [2.0]	

^1^ Statements: (1) Telehealth improves my access to healthcare services. (2) Telehealth saves me time traveling to a hospital or specialist clinic. (3) Telehealth provides for my healthcare need. (4) It was simple to use this system. (5) It was easy to learn to use the system. (6) I believe I could become productive quickly using this system. (7) The way I interact with this system is pleasant. (8) I like using the system. (9) The system is simple and easy to understand. (10) This system is able to do everything I would want it to be able to do. (11) I can easily talk to the clinician using the telehealth system. (12) I can hear the clinician clearly using the telehealth system. (13) I felt I was able to express myself effectively. (14) Using the telehealth system, I can see the clinician as well as if we met in person. (15) I think the visits provided over the telehealth system are the same as in-person visits. (16) Whenever I made a mistake using the system, I could recover easily and quickly. (17) The system gave error messages that clearly told me how to fix problems. (18) I feel comfortable communicating with the clinician using the telehealth system. (19) Telehealth is an acceptable way to receive healthcare services. (20) I would use telehealth services again. (21) Overall, I am satisfied with this telehealth system. ^2^ Numerical value and corresponding statement: 1—strongly disagree, 2—disagree, 3—somewhat disagree, 4—neither agree nor disagree, 5—somewhat agree, 6—agree, 7—strongly agree.

## Data Availability

Data may be available upon reasonable request to the corresponding author.

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
