# Peer review of "Feasibility and First Experiences from an Online Kidney School for Patients with Chronic Kidney Disease"

_ijerph, 2023, doi:10.3390/ijerph20010864_

Round 1

Reviewer 1 Report

Thank you for sharing your work. The OKS seems to have been an important initiative in filling a gap during covid. With regards to the manuscript, I suggest that significant changes need to be done to allow for a comprehensive and sound reporting of this work.

Introduction

More research evidence needs to be presented around the importance of patient education and mHealth in the final stages of CKD. Also show how you incorporated such evidence in the development of the Kidney School and provide available data from its 15 years of running, eg. on its effectiveness, reach, patient satisfaction etc.

OKS- how was it similar or different from F2F? How was it established? What were its goals?

Clearly explain why is this a feasibility study and state its objectives.

Methods

Please consult CONSORT statement extensions on items to report in your manuscript.

Line 53- please further elaborate on what a convergent design entails and how it differs from other types of mixed-methods designs

Line 56- “first experiences”, whose?

2.1 Setting- The paper is not reporting on the physical kidney school, hence this shouldn’t be presented in the methods section.

More details on how the content of the OKS was developed would be useful.

Lines 80-81 incomplete sentence

2.2.2. Please explain on how quality of the data analysis/ trustworthiness was ensured

Line 94 (and throughout)- “inspired”: Perhaps word in a way more appropriate for a research context?

2.2.4 Please present in more detail the revisions and updates that took place.

116-119: More details on the process followed would be useful.

Results

I feel that there is the themes do not seem to reflect the content presented. A richer description, more coherent and backed up with quotes is needed.

For example, under the “quality” theme, the content reads more like concerns/ potential disadvantages of OKS, evolvement of OKS over time and goals of OKS. Under theme two, lines 175-77 read more like a disadvantage than a barrier. Then a reference to the evolvement of OKS over time is again mentioned (180-3). The “Future possibilities” theme content reads as advantages/ benefits, rather than plan/ possibility for the future.

I would suggest reviewing the analysis from earlier stages (grouping of codes).

It would also be helpful to mention whether the various aspects of the themes were unanimous (discussed by all or by most/few FG participants?)

Please review the translated quotes, as several do not seem to make sense in English.

Line 226- earlier in the Results section (line 131) it is stated that 44 participants were invited, here 48. please explain the discrepancy

Line 250-51. A comparison between OKS and PKS is implied here. However, data presented are from the OKS. please explain why or amend

Table 2. Add in a legend the numerical range the questionnaire responses correspond to, so that the reader does not need to go back to the text to find out.

Discussion

Lines 25-26. I don’t feel that this conclusion is clearly supported by the presented data- need to demonstrate this further in the Results section.

Discussion need to be further reviewed following the changes in the result section.

The fact that it is unclear whether patients or relatives responded to the questionnaires (92-94) is a serious limitation. I’m not sure why a screening question (e.g. are you a patient/ relative?) could not be incorporated in the questionnaire pack. Nevertheless, implications for future implementation and assessment of the OKS etc should be discussed.

Author Response

Dear Reviewer

First and foremost, we would like to thank you for the very constructive reviewer feedback and assessments of our manuscript, which we believe we have fully addressed and revised accordingly. 
Comments are addressed below in a Q&A point-by-point manner and changes are marked in the revised version of the manuscript. (Referred to as Action Items) Lines are referring to the lines seen with using the view all changes function in word.
The manuscript has also been read by a native English speaker in order to improve the quality of the language.

Kind regards 

Reviwer 1 Comments

Answers

Action item

 Thank you for sharing your work. The OKS seems to have been an important initiative in filling a gap during covid. With regards to the manuscript, I suggest that significant changes need to be done to allow for a comprehensive and sound reporting of this work.

Thank you for taking the time to review the manuscript. You are absolutely right, the OKS filled an important gap.

Introduction

More research evidence needs to be presented around the importance of patient education and mHealth in the final stages of CKD. Also show how you incorporated such evidence in the development of the Kidney School and provide available data from its 15 years of running, eg. on its effectiveness, reach, patient satisfaction etc.

Thank you for highlighting this. 

Unfortunately, we cannot provide any scientific data regarding the regular kidney school  

The introduction has been extended in order to elaborate further on the evidence of patient education in the final stage of CKD and how mHealth may contribute.

OKS- how was it similar or different from F2F? How was it established? What were its goals?

We agree that the description of the physical kidney school should be should be a part of the introduction.

The description of the physical school has now been moved from setting under methods to the introduction   and the description of the online kidney school and how it was developed has been expanded on and remains under setting in the methods section.

Clearly explain why is this a feasibility study and state its objectives.

Thank you for mentioning this. We agree that the aims of this study were not clearly stated at the end of the introduction.

The ending of the introduction has now been revised to clearly highlight the aims of this study.

Lines: 64-69

Methods

Please consult CONSORT statement extensions on items to report in your manuscript.

Thank you for this great suggestion.

We have now consulted the CONSORT statement for pilot and feasibility studies and included all relevant information throughout the manuscript.

Line 53- please further elaborate on what a convergent design entails and how it differs from other types of mixed-methods designs

In a convergent mixed method design, the qualitative and quantitative data are collected parallel to each other. This is different from both the explanatory and exploratory design where one set of data is collected and analysed leading to and informing the next stage of the mixed methods study.

We have elaborated on that a convergent design entails.

Lines 78-81

Line 56- “first experiences”, whose?

Thanks for pointing this out, we agree that this was not completely clear.

We have now clarified the, sentence.

2.1 Setting- The paper is not reporting on the physical kidney school, hence this shouldn’t be presented in the methods section.

We agree with your suggestion.

The description of the physical kidney school has now been moved from setting under methods to the introduction.

Lines:   47- 51

More details on how the content of the OKS was developed would be useful.

Thank you for pointing this out.

We have now expanded on how the course was converted from a physical to an online kidney school and the theoretical background for the same.

Lines 94 -116

Lines 80-81 incomplete sentence

Thank you for spotting that.

We agree that this sentences needed to be revised and have done so accordingly.

Lines 144-146

2.2.2. Please explain on how quality of the data analysis/ trustworthiness was ensured.

Thank you for pointing this out.

We conducted the steps of the thematic analysis according to Braun and Clark’s methodology and  have amended the manuscript section to be more explicit about how the data were analysed.

Lines: 83-92

Line 94 (and throughout)- “inspired”: Perhaps word in a way more appropriate for a research context?

Thank you for noticing this.

We have change it to

“Adapted from..”

Line 169

2.2.4 Please present in more detail the revisions and updates that took place.

We agree that this was not completely clear

We have elaborate on the process.

Lines 178- 186

116-119: More details on the process followed would be useful.

We agree that this part could be elaborated

The steps of the translation and cultural adaption has now been mentioned.

Lines: 198-203

Results

I feel that there is the themes do not seem to reflect the content presented. A richer description, more coherent and backed up with quotes is needed.

For example, under the “quality” theme, the content reads more like concerns/ potential disadvantages of OKS, evolvement of OKS over time and goals of OKS.

Under theme two, lines 175-77 read more like a disadvantage than a barrier. Then a reference to the evolvement of OKS over time is again mentioned (180-3). The “Future possibilities” theme content reads as advantages/ benefits, rather than plan/ possibility for the future.

I would suggest reviewing the analysis from earlier stages (grouping of codes).

Thank you for your input. Upon reflection, we agree that the themes did indeed need revision.

We have revisited the original codes.

We have revised the themes under one overarching theme with three sub-themes;

Evolvement of the online kidney school overtime

-          Concerns and perceived barriers

-          Facilitators

-          Benefits and future possibilities for the online kidney school.

It would also be helpful to mention whether the various aspects of the themes were unanimous (discussed by all or by most/few FG participants?)

Thank you for pointing this out.

We have amended the analysis accordingly.

Please review the translated quotes, as several do not seem to make sense in English.

The translated quotes have been revised

An English native speaker, fluent in Danish has now read both the Danish and the translated versions of the quotes.

The English translation of the quotes has been reviewed and adjusted where appropriate.

Line 226- earlier in the Results section (line 131) it is stated that 44 participants were invited, here 48. please explain the discrepancy

As stated in the first part of the results section 44 patients of witch 30 participated.

However 48 participants (including both patients and relatives) responded.

We have elaborated that the 48 participants includes both patient and relatives.

Line 250-51. A comparison between OKS and PKS is implied here. However, data presented are from the OKS. please explain why or amend

After the revision of the

This section has now been amended accordingly.

Table 2. Add in a legend the numerical range the questionnaire responses correspond to, so that the reader does not need to go back to the text to find out.

Revisiting table 2 we agree that it was not completely clear.

A legend has been added and the numerical range and corresponding statement as well.

Table2

Discussion

Lines 25-26. I don’t feel that this conclusion is clearly supported by the presented data- need to demonstrate this further in the Results section.

Having amended and clarified our results, we feel that our conclusion is now more supported by the data presented in the results section.

Discussion need to be further reviewed following the changes in the result section.

The discussion has now been revised accordingly

The fact that it is unclear whether patients or relatives responded to the questionnaires (92-94) is a serious limitation. I’m not sure why a screening question (e.g. are you a patient/ relative?) could not be incorporated in the questionnaire pack. Nevertheless, implications for future implementation and assessment of the OKS etc should be discussed.

We agree that we should have incorporated a screening question and it would indeed have improved the study. However, our study was designed rapidly in response to the situation under COvid-19, and therefore was without the benefit of time to reflect over methodological choices.  

We also agree that we should discuss the implications for future implementation and assessment of the OKS.

We have added the context for the design of the study to the methods section and mentioned this again in the limitations section.

We have added a short section on implications for future implementation and assessment of the OKS.

Reviewer 2 Report

1. The authors can add more relevant information in the background section. Example: Data regarding CKD in Denmark and mHealth-related information in this field.

2. Can add a sentence about why telehealth is important in this field.

3. In the method section, the study site and recruitment procedure are unclear.

4. Where the authors conducted the OKS classes needs to specify.

5. Did they authors follow the COREQ guidelines for their qualitative research?

6. In the qualitative data collection section, the authors can mention 1st author or second author, but they used the author's acronym.

7. The authors mentioned they collected data from patients and their relatives. How many were of them?

8. In the KKQ, what the value indicated need to mention.

9. Better to use a validated questionnaire, otherwise it will not be acceptable. If it's not validated, how the authors did validate it?

10. Standard FGD requires 6-8 people per group and at least 2 FGDs for the generalizability of their research. The authors are requested to conduct 1 more FGD and include more HCPs.

11. How the authors could say that OKS was feasible with this small sample?

Author Response

Dear Reviewer

First and foremost, we would like to thank you for the very constructive reviewer feedback and assessments of our manuscript, which we believe we have fully addressed and revised accordingly. 
Comments are addressed below in a Q&A point-by-point manner and changes are marked in the revised version of the manuscript. (Referred to as Action Items) Lines are referring to the lines seen with using the view all changes function in word.
The manuscript has also been read by a native English speaker in order to improve the quality of the language.

Kind regards 

Reviewer 2 Comments

Response

Action item

1. The authors can add more relevant information in the background section. Example: Data regarding CKD in Denmark and mHealth-related information in this field.

Thank you for highlighting this. 

The introduction has been extended in order to elaborate further on data regarding CKD in Denmark,   education in the final stage of CKD and how mHealth may contribute.

2. Can add a sentence about why telehealth is important in this field.

Se response and action item above.

3. In the method section, the study site and recruitment procedure are unclear.

We agree that this was not clear – thank you for pointing this out.

We have elaborated on the study site and the recruitment procedure.

Lines 89-93

4. Where the authors conducted the OKS classes needs to specify.

We have elaborated on this point in the setting and participant section under methods.

The setting and participant section under methods has now been revised to include where the OKS classes were conducted.

Lines 108-120

5. Did they authors follow the COREQ guidelines for their qualitative research?

Yes, we did, however it was not clear.

The sentence regarding COREQ has now been moved from the Institutional review board statement to the method section regarding qualitative research.

Lines 165-166

6. In the qualitative data collection section, the authors can mention 1st author or second author, but they used the author's acronym.

We agree that this would be a better option.

We have now changed the method section accordingly

7. The authors mentioned they collected data from patients and their relatives. How many were of them?

As mentioned in the results, section we collect questionnaires from 48 participants prior to participation and from 24 after participation.

Unfortunately, we were not able to distinguish between patients and relatives. This is of course a major limitation, which we have mentioned under the limitation section of the discussion.  

We have inserted a parenthesis in the beginning of the results section for quantitative results, once again mentioning that the data collected from the participants includes both patients and relatives.

Line:   370

8. In the KKQ, what the value indicated need to mention.

After revisiting the KKQ results section we can see that is was not entirely clear, thank you for mentioning that.

The section has now been clarified.

Lines 375-380

9. Better to use a validated questionnaire, otherwise it will not be acceptable. If it's not validated, how the authors did validate it?

We completely agree that the best option would have been to use an already validated questionnaire, however, there were none available in Danish regarding the usability of telehealth.

Hence we did the best we could under the circumstances and followed the guidelines for translation and cultural adaptation of questionnaires according to ISPOR, as stated in the method section. However, the Danish version has not been validated for internal consistency.

Furthermore, we have mentioned this as a limitation.
We are currently working on validating the questionnaire.

The Method section regarding TUQ has been revised to clarify what we have done to validate and what is still lacking,

Lines: 178-203

Futhermore it mentioned as a limitation and implication for future research.

Lines 108-110 + 129

10. Standard FGD requires 6-8 people per group and at least 2 FGDs for the generalizability of their research. The authors are requested to conduct 1 more FGD and include more HCPs.

We agree that one FGD with only five participants does limit the generalizability of the findings. However, we are unable to conduct more, since all five of the HCPs involved in the OKS were included in this focus group.

The group was neither too heterogeneous nor too homogenous. As the participants in the FGD already know each other, they displayed ease in each other’s company. They demonstrated a high level of interaction and discussion necessary to create empirical data. As the objective of this focus group was to explore the group’s experience with setting up and conducting the OKS, we the authors feel that this method was appropriate despite the fact that only one FGD was performed.

The results section pertaining to the FGD and limitations, which were also addressed in the first version of the manuscript have now been revised slightly in the results section and the limitations section of the discussion.

Lines 223 -231 + 100-104

11. How the authors could say that OKS was feasible with this small sample?

We believe that 30 patients completing the OKS is a fair number for a feasibility study.

This falls well in line with what others have previously reported.
Lewis et al. states that there is no consensus regarding sample size, but that they may vary from 10-75 (2). In an audit of feasibility and pilot studies from United Kingdom, the authors found that the median sample size for feasibility studies were 36 ranging from 10 -300 participants (3).  

The objective of a feasibility study, ours included, is not to asses effectiveness of the intervention, but merely if it is feasible.